# A Case Series of Potential Pediatric Cyanotoxin Exposures Associated with Harmful Algal Blooms in Northwest Ohio

Benjamin W. French [1,†], Rajat Kaul [2,†], Jerrin George [1], Steven T. Haller [1,*], David J. Kennedy [1,*] and Deepa Mukundan [2,*]

[1] Department of Medicine, College of Medicine and Life Sciences, University of Toledo, Toledo, OH 43614, USA; benjamin.french2@rockets.utoledo.edu (B.W.F.); jerrin.george@rockets.utoledo.edu (J.G.)
[2] Department of Pediatrics, College of Medicine and Life Science, University of Toledo, Toledo, OH 43614, USA; rajat.kaul@utoledo.edu
[*] Correspondence: steven.haller@utoledo.edu (S.T.H.); david.kennedy@utoledo.edu (D.J.K.); deepa.mukundan@utoledo.edu (D.M.); Tel.: +1-419-383-6822 (D.J.K.)
[†] These authors contributed equally to this work.

**Abstract:** Cyanobacterial harmful algal blooms (CyanoHABs) are increasing in prevalence and severity in the Great Lakes region, as well as both globally and locally. CyanoHABs have the potential to cause adverse effects on human health due to the production of cyanotoxins from cyanobacteria. Common routes of exposure include recreational exposure (swimming, skiing, and boating), ingestion, and aerosolization of contaminated water sources. Cyanotoxins have been shown to adversely affect several major organ systems contributing to hepatotoxicity, gastrointestinal distress, and pulmonary inflammation. We present three pediatric case reports that coincided with CyanoHABs exposure with a focus on presentation of illness, diagnostic work-up, and treatment of CyanoHAB-related illnesses. Potential cyanotoxin exposure occurred while swimming in the Maumee River and Maumee Bay of Lake Erie in Ohio during the summer months with confirmed CyanoHAB activity. Primary symptoms included generalized macular rash, fever, vomiting, diarrhea, and severe respiratory distress. Significant labs included leukocytosis and elevated C-reactive protein. All patients ultimately recovered with supportive care. Symptoms following potential cyanotoxin exposure coincide with multiple disease states representing an urgent need to develop specific diagnostic tests of exposure.

**Keywords:** cyanoHAB exposure; pediatric case series; menstruation; macular rash; fever; vomiting; diarrhea; respiratory distress; leukocytosis



## 1. Introduction

Cyanobacteria are an ancient and widespread phylum of bacteria, found in nearly every environment on earth. Due to their color, they are often referred to as blue-green algae; while they are not algae, they are capable of photosynthesis, giving them a similar appearance to algae to the naked eye. Due to their photosynthetic nature, cyanobacteria produce chlorophyll a, which contributes to their strong green pigmentation. When these cyanobacteria experience rapid growth, they may begin to start releasing toxins known as cyanotoxins. When large growths of cyanobacteria start producing toxins, they are referred to as cyanobacteria harmful algal blooms, or cyanoHABs. CyanoHABs are driven by eutrophication, including the absolute and relative levels of nitrogen and phosphorus in run-off waters, which is heavily influenced by local agricultural and industrial practices [1].

During CyanoHAB events, the cyanobacteria begin producing a variety of toxins, including microcystins, saxitoxins, anatoxins, and cylindrospermopsins. These CyanoHAB events typically take place in the late summer months, when the waters are warmest, However, several cyanobacteria strains (e.g., *Planktothrix*, *Aphanizomenon*, and *Anabaena/Dolichospermum*) are cold-resistant and thus capable of extending blooms into winter months

and colder climates [2,3]. As CyanoHAB events become more complex and persistent (in some cases, year-round), the window of exposure may be increased, and the number of toxins to which humans and animals may be exposed may increase as multiple cyanobacterial strains produce a variety of cyanotoxins [4,5]. This complexity can result in a wide range of health effects in exposed populations, highlighting the importance of comprehensive monitoring and management strategies. These health effects are far-ranging, and include respiratory, dermatologic, and gastrointestinal, underscoring the need for more research [6].

The family of microcystin toxins is one of the most common cyanotoxins produced in CyanoHABs and includes over 300 congeners [7]. Of these congeners, microcystin-leucine-arginine (MC-LR) is one of the most common and toxic forms. Microcystins, and especially MC-LR, are potent hepatotoxins that enter cells through organic anion transporting polypeptides (OATPs). Microcystins exert their toxic effects largely through their strong inhibition of the serine/threonine protein phosphatase 1 and 2A (PP1 and PP2A) [8]. PP1 and 2A control a huge number of cellular processes (including most steps of the cell cycle) and making up as much as 1% of a cell's total protein [9]. As previously stated, microcystins are known to have toxic effects within the liver, though many of the body's major organ systems can experience these effects due to the expression of key OATPs throughout the body.

Saxitoxin is a paralytical neurotoxin, which also acts as a precursor to several other paralytic shellfish toxins (PSTs). Saxitoxin and its family members function largely by inhibition of the voltage-gated sodium ion channel, though saxitoxin and some of its derivatives can also work to inhibit the potassium and calcium voltage-gated channels [10,11]. Anatoxins, sometimes referred to as the "very fast death factor", are another common form of cyanotoxin. These toxins work by binding to acetylcholine esterase in muscle cells, forcing sodium ion channels to remain open, which then induces uncontrollable muscle contraction. Cylindrospermopsin has a variety of health effects, including gastrointestinal complications, liver inflammation and hemorrhage, pneumonia, and dermatitis. These arise from cylindrospermopsin's inhibition of cytochrome P450 and glutathione-s-transferase [12].

The most common routes of exposure to cyanotoxins are oral/ingestions, inhalation, and dermal contact with contaminated water [6]. Of these, the most highly studied route of exposure is the ingestion of contaminated water or seafood (or cyanobacteria-based food supplements), which can lead to intestinal illness as well as the typical hepatotoxic effects. However, recent studies have shown that cyanotoxins can be aerosolized into water droplets via natural wave motion, showing the need for more studies on the inhalation route of exposure [13,14]. Recently, there is evidence that aerosolized MC-LR induces inflammatory signaling in healthy airway epithelial cells and may increase neutrophilic migration to the airways [14]. Additionally, both epidemiological data and preliminary research indicates some dermatotoxic effects of cyanotoxins, which has been understudied to date [15,16].

Both the frequency and intensity of CyanoHABs have been increasing in recent years, affecting all 48 contiguous states in the U.S. and many countries around the world including New Zealand, China, South Africa, Canada, and Kenya, to name a few [17]. These cyanoHAB events have significant implications for human health, as evidenced by the "Do not drink" advisory in Toledo in 2014 that cut off water access to more than 500,000 residents, or significant mortality at a dialysis center in Caruaru, Brazil in 1996 [6,18]. As CyanoHABs become more frequent, health care professionals must be cognizant of symptoms and various presentations of patients affected by cyanobacterium toxicity. Herein, we present a series of three pediatric cases that coincided with CyanoHAB exposure with a focus on presentation of illness, diagnostic work-up and treatment of CyanoHAB-related illnesses. All three cases occurred in the Western Basin of Lake Erie or in the river that supplies it, and occurred in 2014, 2015, and 2016. The CyanoHAB tracking and intensity data corresponding to the season and year for each of these exposures in the western basin of Lake Erie is shown in Figure 1.

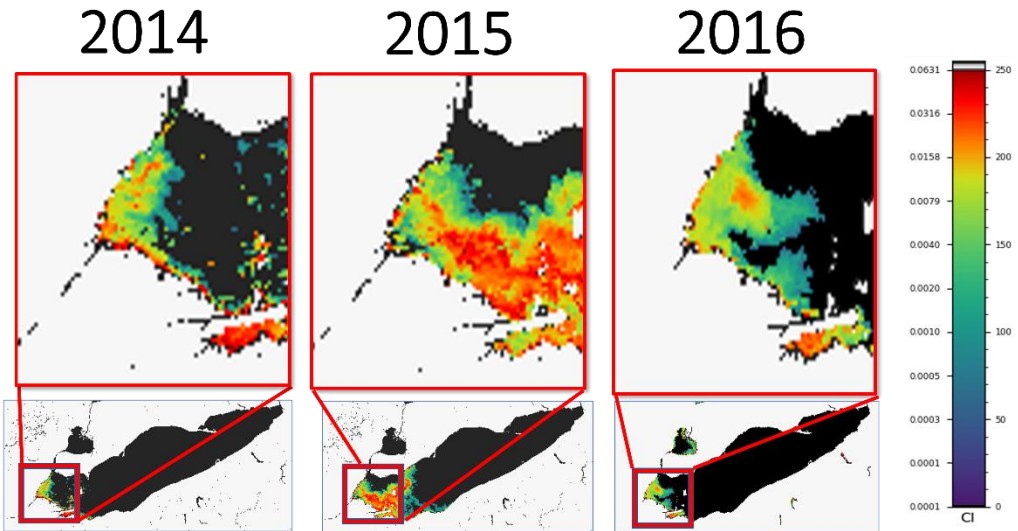

**Figure 1.** MODIS (moderate resolution imaging spectro-radiometer) data of Lake Erie CyanoHAB intensity processed by NOAA National Centers for Coastal Ocean Science. Images of CyanoHAB monitoring presented correlate to the seasons and years of the cases presented below. Method described in Wynne and Stumpf, 2015 [19]. Digital Number (DN) to Cyanobacterial Index value is CI = 10 ^ (3/250 × DN − 4.2). Values of >0.001 (100 DN) is moderate risk, while >0.01 is high risk.

## 2. Case Report

### 2.1. Case 1

Case 1 (2016): A 16-year-old female with no significant past medical history who presented with genital lesions, headache, generalized macular rash and fevers (Table 1). Initially, she was treated as an outpatient for a yeast infection with fluconazole and an over-the-counter cream for vulval itching; however, due to fever and rash associated with worsening of symptoms she was hospitalized. Her CyanoHAB exposure occurred in summer 2016, in the inlet of the Maumee river in Ohio (Figure 1, right column). This exposure occurred 2 weeks prior to admission, while symptoms started 1 week after the exposure. Exposure occurred during swimming in the Maumee river, and the patient did not report ingesting any contaminated waters. The patient was exposed over the course of one day, but did not give an estimation of the time exposed. Work-up was negative for toxin-mediated Staphylococcal disease, including blood culture, swabs from nares, complete blood count, and comprehensive metabolic profile. She was empirically started on antibiotics for Staphylococcal coverage at onset and discontinued as the clinical condition improved and the cultures were back with no growth.

**Table 1.** Patient characteristics summary.

| Patient | Reference Range | Case #1 | Case #2 | Case #3 |
|---|---|---|---|---|
| Age (years) | | 16 | 14 | 7 |
| Sex | | Female | Female | Female |
| CyanoHAB exposure | | Summer 2016 | Summer 2015 | Summer 2014 |
| PMHx | | No significant PMHx | MRSA and depression | Asthma |
| Means of exposure | | Swimming (Maumee river) | Swimming (Maumee Bay) | Swimming (Maumee Bay) |
| Menstruation | | Yes (with tampon) | Yes (with tampon) | No |
| Primary symptoms on admission | | Generalized macular rash, fever, headache, genital ulcers | Generalized macular rash, fever, vomiting, diarrhea, and dehydration | Listless, tachycardia, diminished breath sounds and severe respiratory distress |
| WBC (×1000 per μL) | 5.5–15 × 10⁹ L⁻¹ | 3.6 | 22.5 | 15 |

**Table 1.** *Cont.*

| Patient | Reference Range | Case #1 | Case #2 | Case #3 |
|---|---|---|---|---|
| Band (%) | | N/A | 26% | 20% |
| Hemoglobin (g/dL) | 10.9–14.4 g/dL | 13.2 | 13.3 | 12.2 |
| Platelets (×1000 per μL) | 150–450 | 94 | 247 | 467 |
| AST (U/L) | 0–41 U/L | 192 | 32 | 24 |
| ALT (U/L) | 0–40 U/L | 211 | 15 | 18 |
| Creatinine (mg/dL) | 0.3–1 mg/dL | 0.84 | 1.29 | 0.39 |

Case presentations coinciding with CyanoHAB events. PMHx, past medical history; WBC, white blood count; Band%, band neutrophil concentration; hemoglobin; Platelet, platelet count; AST, aspartate aminotransferase; ALT, alanine aminotransferase.

No further work-up studies were required as the patient's clinical status improved. Notably, the patient was menstruating with tampon use during the exposure. Large CyanoHABs were reported in the area that the patient and her brother were swimming one day after exposure. In the hospital, the patient's labs were significant for thrombocytopenia, transaminitis, and mild leukopenia. Extensive work-up showed no specific etiology and together with her clinical findings and history of exposure, she was presumed to have toxic effects from CyanoHAB exposure and CyanoHAB toxins.

*2.2. Case 2*

Case 2 (2015): A 14-year-old female, with past medical history significant for Methicillin-resistant Staphylococcus aureus (MRSA) skin infection and depression, who presented with 2 days of fever, rash, vomiting, diarrhea, and dehydration. Her CyanoHAB exposure occurred while swimming in Lake Erie's Maumee Bay, in the summer of 2015 (Figure 1, Middle column), which was reported to be 2 days prior to the onset of fevers followed by vomiting and then diarrhea (Table 1). She was also menstruating during exposure with tampon use during her swim in the lake immediately prior to the area closing for human activity due to increased CyanoHABs. No ingestion of contaminated waters was reported. Differential diagnosis on admission included toxic shock syndrome. Her laboratory studies in the hospital were significant for leukocytosis with increased bands, positive for toxic granules in neutrophils, elevated C-reactive protein, and mildly elevated creatinine. Patient was admitted to the pediatric intensive care unit, and had a rapid recovery. Bacterial etiology was ruled out with negative cultures and given the history of CyanoHAB exposure, toxic effects from CyanoHAB toxins was presumed.

*2.3. Case 3*

Case 3 (2014): A 7-year-old female with significant history of poorly controlled asthma presented with decreased responsiveness, tachycardia, diminished breath sounds and severe respiratory distress which required immediate intubation. Her symptoms started immediately after CyanoHAB exposure from swimming in the Maumee Bay in summer 2014 (Figure 1, Left column), and did not improve with albuterol treatments at home. Her chest X-ray showed multiple areas of atelectasis and right lower lobe infiltrate. Her labs were significant for leukocytosis, increased bands (Table 1), and her overall hospital course was lengthy, complicated by steroid-induced myopathy. No other members in the family were exposed, and the patient did not report ingesting any contaminated water. The patient had a prolonged course with respiratory failure. All cultures from blood and respiratory tract were negative for bacteria. She was initially started on empiric antibiotics for about a week for pneumonia. The patient eventually made a full recovery. Based on persistent negative testing for viral or bacterial etiology along with history of CyanoHAB exposure, the patient was diagnosed with presumed toxic effects from CyanoHAB toxins leading to acute respiratory failure.

## 3. Discussion

Here, we present three pediatric patient cases with cyanotoxin exposure that aligned with documented CyanoHAB events in the Western Lake Erie Basin as confirmed by both NOAA MODIS monitoring as well testing performed by the State of Ohio's Environmental Protection Agency and the University of Toledo Lake Erie Center [19–21]. The patients were females between the ages of 7 and 16, presenting with symptoms such as generalized macular rash, fever, vomiting, diarrhea, and severe respiratory distress, among others. All three were exposed to waters that were contaminated with cyanotoxin-producing bacteria around the same time as onset of symptoms. In each case, symptoms resolved with supportive care, and patients recovered quickly. Notably, two of the cases involved patients who were menstruating with tampon use. Both patients presented with a generalized rash, along with signs of organ dysfunction (case 1 with transaminitis and case 2 with elevated creatinine). Notably, the male sibling of case 1 had also developed symptoms after recreational exposure, but those symptoms self-resolved quickly and did not require any medical intervention. One may postulate that there was a higher and potentially prolonged level of toxic exposure which may have contributed to more severe symptoms. The mechanism of action may be similar to women that develop toxic shock syndrome secondary to tampon retention. Further, a male sibling of one of our patients had concurrent exposure, which resulted in mild symptoms that self-resolved, suggesting that menstruation, and possibly having a tampon, may play a role in the degree of CyanoHAB-related exposure and subsequent illness. It is possible that the use of tampons while exposure was occurring allowed the material to absorb contaminated waters, giving a prolonged exposure relative to those who were exposed without the use of tampons.

Currently, the diagnosis of cyanotoxin exposure and related illness is a diagnosis of exclusion. The World Health Organization (WHO) is responsible for the creation and maintenance of the International Classification of Diseases (ICD) classification system to serve as a key method for identifying health trends and statistics globally and is the international standard for reporting mortality, morbidity and other conditions affecting health including diagnoses, symptoms and procedures recorded in conjunction with hospital care. The ICD-10 cm (International Classification of Diseases, Tenth Revision, Clinical Modification) in the United States contains specific codes for both "Contact with and (suspected) exposure to harmful algae and algae toxins" (ICD-10 cm Code Z77.121) and "Toxic effect harmful algae and algae toxins" (ICD-10 cm Code T65.82). Lack of awareness of both CyanoHABs-specific ICD codes and CyanoHABs in general, may lead to underreporting of exposure and toxicity events. The Centers for Disease Control and Prevention (CDC) recommends the use of these codes in diagnosing and recording CyanoHAB-related exposure and illnesses (Table 2).

CyanoHABs are a growing public health concern, and key knowledge gaps in cyanotoxin research need to be addressed. Though the liver is a key target for cyanotoxins such as microcystin, work from our lab has shown that merely monitoring aspartate aminotransferase (AST) or alanine aminotransferase (ALT) levels may be insufficient for diagnosis, requiring other methods of detecting damage to organ systems such as the liver Additionally, work from our lab and others show that microcystin impacts the kidneys and gut, and may also work as a cardio [22,23] and neurotoxin [24]. Beyond microcystin, other cyanotoxins (e.g., saxitoxin, anatoxin, and cylindrospermopsin, among others and other cyanobacterial metabolites) affect a variety of organs and organ systems. While microcystins in tissues can be detected using enzyme-linked immunosorbent assays (ELISA), protein phosphatase inhibition assays, and Lemieux oxidation; none of these methods are capable of differentiating between different congeners and metabolites of microcystin [7].

**Table 2.** CyanoHAB references for healthcare and poison control professionals.

| Agency | Resource and Description |
|---|---|
| World Health Organization (WHO) International Classification of Diseases (ICD) | ICD-10 cm codes recommended by the Centers for Disease Control and Prevention (CDC) for use in diagnosing and recording CyanoHAB-related exposure and illnesses: **Z77.121:** Contact with and (suspected) exposure to harmful algae and algae toxins **T65.82:** Toxic effect harmful algae and algae toxins WHO guidelines: https://www.who.int/publications/m/item/toxic-cyanobacteria-in-water---second-edition (accessed on 1 November 2023). |
| Centers for Disease Control and Prevention (CDC) | Harmful Algal Bloom-Associated Illness Fact Sheets and Reference Cards (Physician Reference, Cyanobacteria FAQ, Facts about Cyanobacterial Blooms for Poison Center Professionals) https://www.cdc.gov/habs/materials/factsheets.html (accessed on 1 November 2023). |
| One Health Harmful Algal Bloom Reporting System (OHHABS) | Infosheet on OHHABS Reporting, including summary information on CyanoHABs and those who may be affected and helping https://www.cdc.gov/habs/pdf/ohhabs-reporting-flow-diagram-508.pdf (accessed on 1 November 2023). Factsheet on CyanoHABs from OHHABS, providing digestible information on CyanoHABs, reporting, surveillance, and current systems in place. https://www.cdc.gov/habs/pdf/ohhabs-fact-sheet.pdf (accessed on1 November 2023). User Resources for the: One Health Harmful Algal Bloom System (OHHABS) and National Outbreak Reporting System (NORS) https://www.cdc.gov/habs/pdf/ohhabs-fact-sheet.pdf (accessed on 1 November 2023). |
| Great Lakes HABs Collaborative | Information on Harmful Algal Blooms and Human Health Effects https://www.glc.org/wp-content/uploads/HABS-FactSheet-Chronic-Health-202205.pdf (accessed on 1 November 2023). Easily digestible text and graphics for the inhalation risks of CyanoHAB toxins https://www.glc.org/wp-content/uploads/HABS-FactSheet-Toxins-in-Air-202205.pdf (accessed on 1 November 2023). |
| Ohio Department of Health | Reporting Human Illness from Recreational CyanoHAB Exposure https://odh.ohio.gov/know-our-programs/harmful-algal-blooms/forms/habs-illness-form-recreation (accessed on 1 November 2023). Reporting Human Illness from Ingestion of CyanoHAB Contaminated Waters https://odh.ohio.gov/know-our-programs/harmful-algal-blooms/forms/habs-human-illness-form-drinking (accessed on 1 November 2023). |

Harmful Algal Bloom references and resources for physicians and poison control professionals. Resources come from health and regulatory agencies listed in the table.

At the time patients were hospitalized, no lab testing for cyanotoxins was available. The patients in this case series had negative lab findings for any other source, recovered with supportive care, and all had exposure to a harmful algae bloom shortly before symptom onset. However, clinical testing would be greatly advantageous. Our preliminary work indicates that high-resolution Mass Spectrometry (MS) and Matrix-assisted laser desorption/ionization mass spectrometry (MALDI-MS) imaging could be useful for closing some of these knowledge gap [7,25]. MS is a powerful tool that could potentially be used for the detection of some cyanotoxins and their metabolites from patient samples [7]. Thanks

to the unparalleled specificity and sensitivity of MS-based technologies, their inclusion in a standardized toxicological assessments would be beneficial by allowing more definitive serologic assessments, thus improving accuracy of a differential diagnosis. This would both allow healthcare providers to diagnose what specific toxin(s) may be responsible for illness, as well as provide appropriate supportive care. Our lab has also helped to develop techniques that reveal the spatial distribution of some cyanotoxins in tissue, and can even reliably detect the concentration gradient throughout a tissue section. While these are still in the early stages of development, they show promising results for the detection of microcystin from plasma, urine, as well as the presence and localization of microcystins in tissue [7,25]. Practical and reliable detection of cyanotoxins, along with pathologic thresholds, would greatly assist diagnosis and targeted treatment of patients exposed to cyanotoxins. By being able to test for different cyanotoxins, and knowing at what levels they become dangerous, we would be able to better screen patients for short-term and long-term exposure risks.

The prevalence and persistence of CyanoHABs are increasing globally, raising the likelihood that more people will be at risk for cyanotoxin exposure and illness. Additionally, work from our lab has shown that several common comorbidities affecting the liver (non-alcoholic fatty liver disease, or NAFLD) [26–29], gut (colitis/irritable bowel disease or IBD) [30,31], and airways (asthma) [14] may increase susceptibility to cyanotoxins such as microcystin. These conditions are notable as they are already increasingly common As the incidence and prevalence of diseases such as non-alcoholic fatty liver disease, inflammatory bowel disease, and asthma increase, this may have profound impacts on the health of at-risk populations who are exposed to CyanoHABs.

## 4. Conclusions

While the patients in this series recovered with minimal complications, we believe that this series highlights important issues around CyanoHABs and cyanotoxins. Because specific diagnostic and therapeutic options for CyanoHAB related exposure and illness are lacking, we have an incomplete understanding of the extent and occurrence of the cyanotoxin events. Thus, it is important that healthcare providers and public health officials are vigilant in recording and tracking exposure events and related illness so that a more complete clinical picture of patient symptoms and outcomes can inform advancements in preventative, diagnostic and therapeutic strategies. Toward this end, Table 2 highlights several useful resources for healthcare providers and public health officials that can be used for diagnosing and recording CyanoHAB-related exposure and illnesses as well as for education of the public about the health risks of CyanoHAB-related exposures. Some of these resources are state-specific, so it is important that physicians understand their own state's reporting protocols and systems. Also included in Table 2 are resources with lay terms and graphics for public education on CyanoHABs and their risks.

**Author Contributions:** Writing—original draft, R.K. and D.M.; writing—reviewing and editing, B.W.F., R.K., J.G., D.J.K., S.T.H. and D.M.; investigation, R.K. and D.M.; visualization, B.W.F., J.G., D.J.K. and S.T.H.; supervision, R.K., D.J.K., S.T.H. and D.M.; data curation, R.K. and B.W.F. All authors have read and agreed to the published version of the manuscript.

**Funding:** This research was funded by the Harmful Algal Bloom Research Initiative grants from the Ohio Department of Higher Education and the David and Helen Boone Foundation Research Fund.

**Institutional Review Board Statement:** The Institutional Review Board of ProMedica approved this case series as not being considered human subjects' research (IRB#23-111).

**Informed Consent Statement:** In accordance with the Institutional Review Board recommendation who also determined that the proposed case report/case series is not considered human subjects' research, patient consent was not required as the data did not contain any identifiable information (the list of what constitutes identifiable information can be found here) and the data did not go beyond our involvement of the care plan.

**Data Availability Statement:** All relevant data are available in Table 1.

**Acknowledgments:** The authors gratefully acknowledge Margaret Hoogland for her expert assistance in manuscript editing.

**Conflicts of Interest:** The authors declare no conflict of interest.

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
