# Peer review of "A Case Series of Potential Pediatric Cyanotoxin Exposures Associated with Harmful Algal Blooms in Northwest Ohio"

_2036-7449, doi:10.3390/idr15060065_

Round 1
Reviewer 1 Report
Comments and Suggestions for Authors
The authors provide a description of three suspected cases series of toxic cyanobacteria exposures.
Comments
1) The authors state no IRB review was needed, but how were medical files and records accessed for the summary? How were the cases identified?
2) There is no direct evidence that the cases were actually exposed to cyanotoxins. The NOAA map provided shows annual risk for cyanobacteria blooms, and apparently neither the water nor the patients were tested for microcystin or other toxic alga. This needs to be discussed and mentioned as a limitation
3) Line 113: "Large CyanoHABs were reported in the area..." this statement needs a reference to support it
4) In several places the authors seem to imply some importance of the fact that two of the cases were menstruating but they do not clearly discuss why. Can the implication of this please be made more clear? Is it assumed to be an exposure route, or do they believe it may make them more vulnerable for some reason? Please clarify
5) Were the cases diagnosed with the ICD CM code for toxic algae exposure? If not, why not?
6) Line 210- incomplete sentence "Because specific diagnostic and therapeutic options for CyanoHAB related exposure and illness are lacking, we have an incomplete clinical picture of the extent of"
7) The authors should also note that the WHO has released the 2nd edition of toxic cyanobacteria in water: https://www.who.int/publications/m/item/toxic-cyanobacteria-in-water---second-edition and that the US EPA has developed drinking water and swimming advisories for cyanotoxin
8) it may be beyond the scope of the paper but I was surprised to see that the authors do not mention the potential risks through other water exposures, and infamous HAB events such as the deaths due to liver failure in Brazil when microcystin contaminated water was used in dialysis, and the 2014 bloom that forced Toledo to shut down it's drinking water
Comments on the Quality of English Language
English is fine except typo noted above
Reviewer 2 Report
Comments and Suggestions for Authors
This is a good report. I would recommend to the authors to expand the paragraph in lines 190-202 and discuss in 2-3 lines why MS analysis should be included in standardized toxicology screening, or at least as a standardized method. I know of several cases in Wisconsin where fentanyl-mixtures were the primary suspect for an overdose presentation, but there were a number of inconsistencies, especially with only residual amounts of illicit drugs in the patients. There was a suspicion of potential HAB toxicity because many of these homeless patients abide along the shores of these eutrophying lakes. It was discussed about where to send samples for analysis (e.g., NOAA Charleston lab), but that idea was abandoned because progression of the disease and costs. Anyhow, it may be that sooner or later toxicological screening may need to be able to differentiate a HAB toxicity from suspected fentanyl-mixture ODs.
Reference 13, I think you left out the name of the journal, Environ. Sci. Technol.
Reviewer 3 Report
Comments and Suggestions for Authors
Manuscript A Case Series of Potential Pediatric Cyanotoxin Exposures Associated with Harmful Algal Blooms in Northwest Ohio presents an interesting presentation of three pediatric case-reports that are connected with cyanobacterial exposure. Such papers are not frequent enough and are important for the scientific community, but also more widely. For that reason, there is significant value in this paper; however, it is necessary to make certain revisions before publication.
In general, manuscript is promising; however the main complaint is that it is incomplete. It should be more informative, to get a complete picture of the state of the water body, exposure, intoxication, connections between them, as well as presentation and emphasis of the problem in general, through a more detailed presentation of the literature and more detailed discussions. Suggestions and comments are listed below in more detail:
ABSTRACT
20 treatment of CyanoHAB related illnesses was not provided in the text, please add it, or omit this part
INTRODUCTION
36-40 Cyanobacterial harmful algal blooms should be explained better and clearer
50 health effects in exposed populations should be highlighted, more information on the health consequences should be added in the text to emphasize the importance of this issue
55 Organic anion transporting polypeptides
76 even ingestion of cyanobacterial based food supplements
90 treatment of CyanoHAB related illnesses was not specified in the case descriptions, it would be significant to add that information
Figure 1 should be explained in more detail, since it is the only data on the presence of cyanobacteria. Are there any previous publications investigating cyanobacteria and cyanotoxins at those sites? If they exist, it would be important to include them in the text. Also, mark the exact points on the maps where exposures occurred.
Case Report
Case 1
More information about the exposure should be provided, such as how the exposure occurred, for how long (days, hours). Give more information about the brother, his exposure and symptoms. Were there other symptoms, was the rash localized elsewhere? What about work-up? Were other analyzes done, a vaginal swab taken, to eliminate other pathogens, e.g. bacteria, viruses,...? Which therapy was eventually used and showed results?
Case 2
Explain MRSA.
Again give more details about exposure, how long and how, did she swallow water while swimming? Is there more information about analyzes and what therapy was applied?
Case 3
Also offer more detailed information related to the case. Were there other exposed people, family members, did they also have symptoms? Indicate treatment and applied therapy.
Table 1 Add reference values (from-to) in the new column for laboratory findings
Discussion
Discuss cases more, compare symptoms, laboratory findings, find regularities, give explanations, and draw common conclusions.
163 explain how you think menstruation, or more precisely the use of tampons, affects the exposure and appearance of symptoms
182 specify which diagnostic parameters should be required
185 (e.g. saxitoxin, anatoxin, and cylindrospermopsin) in addition to mentioned cyanotoxins, there are other cyanotoxins and metabolites of cyanobacteria that can also be harmful
186 systems8 ?
190 it's a pity that those analyzes were not done in the mentioned cases
201 would be greatly assist - delete be
204-207 elaborate in more detail
Table 2 State the basic guidelines, useful information from the sources listed in table 2, so that the reader does not have to find sources and read everything, but serve the readers the most important and useful information related to the problem, thus the paper itself gains value and presents what is necessary - a synthesis of important and useful information for recognition, diagnosis, analysis and therapy in case of exposure and intoxication by cyanobacterial metabolites, make some of these guidelines yours missing Conclusion
Add Conclusion
Round 2
Reviewer 3 Report
Comments and Suggestions for Authors
Most of the suggestions have been included, but there are still a few questions and corrections left.
Why was the information about the brother from the case one omitted? That was a significant information.
Discussion page 7 (e.g. saxitoxin, anatoxin, and cylindrospermopsin, among others and other cyanobacterial metabolites)
The first sentence in the conclusion is not ok.
The conclusion and table 2 should be more informative.
Comments on the Quality of English LanguageGo over the paper because there are some typing errors.
